# Stochastic Bandits with Context Distributions

**Johannes Kirschner**
Department of Computer Science
ETH Zurich
jkirschner@inf.ethz.ch

**Andreas Krause**
Department of Computer Science
ETH Zurich
krausea@ethz.ch

## Abstract

We introduce a stochastic contextual bandit model where at each time step the environment chooses a distribution over a context set and samples the context from this distribution. The learner observes only the context distribution while the exact context realization remains hidden. This allows for a broad range of applications where the context is stochastic or when the learner needs to predict the context. We leverage the UCB algorithm to this setting and show that it achieves an order-optimal high-probability bound on the cumulative regret for linear and kernelized reward functions. Our results strictly generalize previous work in the sense that both our model and the algorithm reduce to the standard setting when the environment chooses only Dirac delta distributions and therefore provides the exact context to the learner. We further analyze a variant where the learner observes the realized context after choosing the action. Finally, we demonstrate the proposed method on synthetic and real-world datasets.

## 1 Introduction

In the contextual bandit model a learner interacts with an environment in several rounds. At the beginning of each round, the environment provides a context, and in turn, the learner chooses an action which leads to an a priori unknown reward. The learner's goal is to choose actions that maximize the cumulative reward, and eventually compete with the best mapping from context observations to actions. This model creates a dilemma of *exploration and exploitation*, as the learner needs to balance exploratory actions to estimate the environment's reward function, and exploitative actions that maximize the total return. Contextual bandit algorithms have been successfully used in many applications, including online advertisement, recommender systems and experimental design.

The contextual bandit model, as usually studied in the literature, assumes that the context is observed *exactly*. This is not always the case in applications, for instance, when the context is itself a noisy measurement or a forecasting mechanism. An example of such a context could be a weather or stock market prediction. In other cases such as recommender systems, privacy constraints can restrict access to certain user features, but instead we might be able to infer a distribution over those. To allow for uncertainty in the context, we consider a setting where the environment provides a *distribution over the context set*. The exact context is assumed to be a sample from this distribution, but remains hidden from the learner. Such a model, to the best of our knowledge, has not been discussed in the literature before. Not knowing the context realization makes the learning problem more difficult, because the learner needs to estimate the reward function from noisy observations and without knowing the exact context that generated the reward. Our setting recovers the classical contextual bandit setting when the context distribution is a Dirac delta distribution. We also analyze a natural variant of the problem, where the exact context is observed *after* the player has chosen the action. This allows for different applications, where at the time of decision the context needs to be predicted (e.g. weather conditions), but when the reward is obtained, the exact context can be measured.

We focus on the setting where the reward function is linear in terms of action-context feature vectors. For this case, we leverage the UCB algorithm on a specifically designed bandit instance *without* feature uncertainty to recover an $\mathcal{O}(d\sqrt{T})$ high-probability bound on the cumulative regret. Our analysis includes a practical variant of the algorithm that requires only sampling access to the context distributions provided by the environment. We also extend our results to the kernelized setting, where the reward function is contained in a known reproducing kernel Hilbert space (RKHS). For this case, we highlight an interesting connection to distributional risk minimization and we show that the natural estimator for the reward function is based on so-called kernel mean embeddings. We discuss related work in Section 6.

## 2 Stochastic Bandits with Context Distributions

We formally define the setting of *stochastic bandits with context distributions* as outlined in the introduction. Let $\mathcal{X}$ be a set of actions and $\mathcal{C}$ a context set. The environment is defined by a fixed, but unknown reward function $f : \mathcal{X} \times \mathcal{C} \to \mathbb{R}$. At iteration $t \in \mathbb{N}$, the environment chooses a distribution $\mu_t \in \mathcal{P}(\mathcal{C})$ over the context set and samples a context realization $c_t \sim \mu_t$. The learner observes only $\mu_t$ but not $c_t$, and then chooses an action $x_t \in \mathcal{X}$. We allow that an *adaptive adversary* chooses the context distribution, that is $\mu_t$ may in an arbitrary way depend on previous choices of the learner up to time $t$. Given the learner's choice $x_t$, the environment provides a reward $y_t = f(x_t, c_t) + \epsilon_t$, where $\epsilon_t$ is $\sigma$-subgaussian, additive noise. The learner's goal is to maximize the cumulative reward $\sum_{t=1}^{T} f(x_t, c_t)$, or equivalently, minimize the cumulative regret

$$\mathcal{R}_T = \sum_{t=1}^{T} f(x_t^*, c_t) - f(x_t, c_t) \tag{1}$$

where $x_t^* = \arg\max_{x \in \mathcal{X}} \mathbb{E}_{c \sim \mu_t}[f(x, c)]$ is the best action provided that we know $f$ and $\mu_t$, but not $c_t$. Note that this way, we compete with the best possible mapping $\pi^* : \mathcal{P}(\mathcal{C}) \to \mathcal{X}$ from the observed context distribution to actions, that maximizes the expected reward $\sum_{t=1}^{T} \mathbb{E}_{c_t \sim \mu_t}[f(\pi^*(\mu_t), c_t)|\mathcal{F}_{t-1}, \mu_t]$ where $\mathcal{F}_t = \{(x_s, \mu_s, y_s)\}_{s=1}^{t}$ is the filtration that contains all information available at the end of round $t$. It is natural to ask if it is possible to compete with the stronger baseline that chooses actions given the context realization $c_t$, i.e. $\tilde{x}_t^* = \arg\max_{x \in \mathcal{X}} f(x, c_t)$. While this can be possible in special cases, a simple example shows, that in general the learner would suffer $\Omega(T)$ regret. In particular, assume that $c_t \sim \text{Bernoulli}(0.6)$, and $\mathcal{X} = \{0, 1\}$. Let $f(0, c) = c$ and $f(1, c) = 1 - c$. Clearly, any policy that does not know the realizations $c_t$, must have $\Omega(T)$ regret when competing against $\tilde{x}_t^*$.

From now on, we focus on linearly parameterized reward functions $f(x, c) = \phi_{x,c}^\top \theta$ with given feature vectors $\phi_{x,c} \in \mathbb{R}^d$ for $x \in \mathcal{X}$ and $c \in \mathcal{C}$, and unknown parameter $\theta \in \mathbb{R}^d$. This setup is commonly referred to as the *linear bandit* setting. For the analysis we require standard boundedness assumptions $\|\phi_{x,c}\|_2 \le 1$ and $\|\theta\|_2 \le 1$ that we set to 1 for the sake of simplicity. In Section 4.2, we further consider a variant of the problem, where the learner observes $c_t$ *after* taking the decision $x_t$. This simplifies the estimation problem, because we have data $\{(x_t, c_t, y_t)\}$ with exact context $c_t$ available, just like in the standard setting. The exploration problem however remains subtle as at the time of decision the learner still knows only $\mu_t$ and not $c_t$. In Section 4.3 we extend our algorithm and analysis to *kernelized bandits* where $f \in \mathcal{H}$ is contained in a reproducing kernel Hilbert space $\mathcal{H}$.

## 3 Background

We briefly review standard results from the linear contextual bandit literature and the upper confidence bound (UCB) algorithm that we built on later (Abbasi-Yadkori et al., 2011). The *linear contextual bandit* setting can be defined as a special case of our setup, where the choice of $\mu_t$ is restricted to Dirac delta distributions $\mu_t = \delta_{c_t}$, and therefore the learner knows beforehand the exact context which is used to generate the reward. In an equivalent formulation, the environment provides at time $t$ a set of action-context feature vectors $\Psi_t = \{\phi_{x,c_t} : x \in \mathcal{X}\} \subset \mathbb{R}^d$ and the algorithm chooses an action $x_t$ with corresponding features $\phi_t := \phi_{x_t,c_t} \in \Psi_t$. We emphasize that in this formulation the context $c_t$ is extraneous to the algorithm, and everything can be defined in terms of the time-varying action-feature sets $\Psi_t$. As before, the learner obtains a noisy reward observation $y_t = \phi_t^\top \theta + \epsilon_t$ where $\epsilon_t$ is conditionally $\rho$-subgaussian with *variance proxy* $\rho$, i.e.

$$\forall \lambda \in \mathbb{R}, \qquad \mathbb{E}[e^{\lambda \epsilon_t} | \mathcal{F}_{t-1}, \phi_t] \le \exp(\lambda^2 \rho^2 / 2) \,.$$

Also here, the standard objective is to minimize the cumulative regret $\mathcal{R}_T = \sum_{t=1}^{T} \phi_t^{*\top} \theta - \phi_t^{\top} \theta$ where $\phi_t^* = \arg\max_{\phi \in \Psi_t} \phi^{\top} \theta$ is the feature vector of the best action at time $t$.

To define the UCB algorithm, we make use of the confidence sets derived by Abbasi-Yadkori et al. (2011) for online least square regression. At the end of round $t$, the algorithm has adaptively collected data $\{(\phi_1, y_1), \ldots, (\phi_t, y_t)\}$ that we use to compute the regularized least squares estimate $\hat{\theta}_t = \arg\min_{\theta' \in \mathbb{R}^d} \sum_{s=1}^{t} (\phi_s^{\top} \theta' - y_s)^2 + \lambda \|\theta'\|_2^2$ with $\lambda > 0$. We denote the closed form solution by $\hat{\theta}_t = V_t^{-1} \sum_{s=1}^{t} \phi_s y_s$ with $V_t = V_{t-1} + \phi_t \phi_t^{\top}$, $V_0 = \lambda \mathbf{I}_d$ and $\mathbf{I}_d \in \mathbb{R}^{d \times d}$ is the identity matrix.

**Lemma 1** (Abbasi-Yadkori et al. (2011)). *For any stochastic sequence $\{(\phi_t, y_t)\}_t$ and estimator $\hat{\theta}_t$ as defined above for $\rho$-subgaussian observations, with probability at least $1 - \delta$, at any time $t \in \mathbb{N}$,*

$$\|\theta - \hat{\theta}_t\|_{V_t} \leq \beta_t \qquad where \; \beta_t = \beta_t(\rho, \delta) = \rho\sqrt{2 \log\left(\frac{\det(V_t)^{1/2}}{\delta \det(V_0)^{1/2}}\right)} + \lambda^{1/2}\|\theta\|_2 \,.$$

Note that the size of the confidence set depends the variance proxy $\rho$, which will be important in the following. In each round $t + 1$, the UCB algorithm chooses an action $\phi_{t+1}$, that maximizes an upper confidence bound on the reward,

$$\phi_{t+1} := \arg\max_{\phi \in \Psi_{t+1}} \phi^{\top} \hat{\theta}_t + \beta_t \|\phi\|_{V_t^{-1}} \,.$$

The following result shows that the UCB policy achieves sublinear regret (Dani et al., 2008; Abbasi-Yadkori et al., 2011).

**Lemma 2.** *In the standard contextual bandit setting with $\rho$-subgaussian observation noise, the regret of the UCB policy with $\beta_t = \beta_t(\rho, \delta)$ is bounded with probability $1 - \delta$ by*

$$\mathcal{R}_T^{UCB} \leq \beta_T \sqrt{8T \log\left(\frac{\det V_T}{\det V_0}\right)}$$

The data-dependent terms can be further upper-bounded to obtain $\mathcal{R}_T^{UCB} \leq \tilde{\mathcal{O}}(d\sqrt{T})$ up to logarithmic factors in $T$ (Abbasi-Yadkori et al., 2011, Theorem 3). A matching lower bound is given by Dani et al. (2008, Theorem 3).

## 4 UCB with Context Distributions

In our setting, where we only observe a context distribution $\mu_t$ (e.g. a weather prediction) instead of the context $c_t$ (e.g. realized weather conditions), also the features $\phi_{x,c_t}$ (e.g. the last layer of a neural network that models the reward $f(x, c) = \phi_{x,c_t}^{\top} \theta$) are uncertain. We propose an approach that transforms the problem such that we can directly use a contextual bandit algorithm as for the standard setting. Given the distribution $\mu_t$, we define a new set of feature vectors $\Psi_t = \{\bar{\psi}_{x,\mu_t} : x \in \mathcal{X}\}$, where we denote by $\bar{\psi}_{x,\mu_t} = \mathbb{E}_{c \sim \mu_t}[\phi_{x,c} | \mathcal{F}_{t-1}, \mu_t]$ the expected feature vector of action $x$ under $\mu_t$. Each feature $\bar{\psi}_{x,\mu_t}$ corresponds to exactly one action $x \in \mathcal{X}$, so we can use $\Psi_t$ as feature context set at time $t$ and use the UCB algorithm to choose an action $x_t$. The choice of the UCB algorithm here is only for the sake of the analysis, but any other algorithm that works in the linear contextual bandit setting can be used. The complete algorithm is summarized in Algorithm 1. We compute the UCB action $x_t$ with corresponding expected features $\psi_t := \bar{\psi}_{x_t,t} \in \Psi_t$, and the learner provides $x_t$ to the environment. We then proceed and use the reward observation $y_t$ to update the least squares estimate. That this is a sensible approach is not immediate, because $y_t$ is a noisy observation of $\phi_{x_t,c_t}^{\top} \theta$, whereas UCB expects the reward $\psi_t^{\top} \theta$. We address this issue by constructing the feature set $\Psi_t$ in such a way, that $y_t$ acts as unbiased observation also for the action choice $\psi_t$. As computing exact expectations can be difficult and in applications often only sampling access of $\mu_t$ is possible, we also analyze a variant of Algorithm 1 where we use finite sample averages $\tilde{\psi}_{x,\mu_t} = \frac{1}{L} \sum_{l=1}^{L} \phi_{x,\tilde{c}_l}$ for $L \in \mathbb{N}$ i.i.d. samples $\tilde{c}_l \sim \mu_t$ instead of the expected features $\bar{\psi}_{x,\mu}$. The corresponding feature set is $\tilde{\Psi}_t = \{\tilde{\psi}_{x,\mu_t} : x \in \mathcal{X}\}$. For both variants of the algorithm we show the following regret bound.

---

**Algorithm 1** UCB for linear stochastic bandits with context distributions

---

Initialize $\hat{\theta} = 0 \in \mathbb{R}^d$, $V_0 = \lambda \mathbf{I} \in \mathbb{R}^{d \times d}$
**For** step $t = 1, 2, \ldots, T$:
     *Environment* chooses $\mu_t \in \mathcal{P}(\mathcal{C})$                       *// context distribution*
     *Learner* observes $\mu_t$

     Set $\Psi_t = \{\bar{\psi}_{x,\mu_t} : x \in \mathcal{X}\}$ with $\bar{\psi}_{x,\mu_t} := \mathbb{E}_{c \sim \mu_t}[\phi_{x,c}]$      *// variant 1, expected version*
     Alternatively, sample $c_{t,1}, \ldots, c_{t,L}$ for $L = t$,          *// variant 2, sampled version*
     Set $\Psi_t = \{\tilde{\psi}_{x,\mu_t} : x \in \mathcal{X}\}$ with $\tilde{\psi}_{x,\mu_t} = \frac{1}{L}\sum_{l=1}^{L} \phi_{x,\tilde{c}_l}$

     Run UCB step with $\Psi_t$ as context set                              *// reduction*
     Choose action $x_t = \arg\max_{\psi_{x,\mu_t} \in \Psi_t} \psi_{x,\mu_t}^\top \hat{\theta}_{t-1} + \beta_t \|\psi_{x,\mu_t}\|_{V_{t-1}^{-1}}$     *// UCB action*

     *Environment* provides $y_t = \phi_{x_t,c_t}^\top \theta + \epsilon$ where $c_t \sim \mu_t$      *// reward observation*
     Update $V_t = V_{t-1} + \psi_{x_s,\mu_s}\psi_{x_s,\mu_s}^\top$, $\hat{\theta}_t = V_t^{-1}\sum_{s=1}^{t} \psi_{x_s,\mu_s} y_s$     *// least-squares update*

---

**Theorem 1.** *The regret of Algorithm 1 with expected feature set $\Psi_t$ and $\beta_t = \beta_t(\sqrt{4 + \sigma^2}, \delta/2)$ is bounded at time $T$ with probability at least $1 - \delta$ by*

$$\mathcal{R}_T \leq \beta_T \sqrt{8T \log\left(\frac{\det V_T}{\det V_0}\right)} + 4\sqrt{2T \log\frac{4}{\delta}} \, .$$

*Further, for finite action sets $\mathcal{X}$, if the algorithm uses sampled feature sets $\tilde{\Psi}_t$ with $L = t$ and $\beta_t = \tilde{\beta}_t$ as defined in* (11)*, Appendix A.2, then with probability at least $1 - \delta$,*

$$\mathcal{R}_T \leq \tilde{\beta}_T \sqrt{8T \log\left(\frac{\det V_T}{\det V_0}\right)} + 4\sqrt{2T \log\frac{2|\mathcal{X}|\pi T}{3\delta}} \, .$$

As before, one can further upper bound the data-dependent terms to obtain an overall regret bound of order $\mathcal{R}_T \leq \tilde{\mathcal{O}}(d\sqrt{T})$, see (Abbasi-Yadkori et al., 2011, Theorem 2). With iterative updates of the least-squares estimator, the per step computational complexity is $\mathcal{O}(Ld^2|\mathcal{X}|)$ if the UCB action is computed by a simple enumeration over all actions.

## 4.1    Regret analysis: Proof of Theorem 1

Recall that $x_t$ is the action that the UCB algorithm selects at time $t$, $\psi_t$ the corresponding feature vector in $\Psi_t$ and we define $\psi_t^* = \arg\max_{\psi \in \Psi_t} \psi^\top \theta$. We show that the regret $\mathcal{R}_T$ is bounded in terms of the regret $\mathcal{R}_T^{UCB} := \sum_{t=1}^{T} \psi_t^{*\top}\theta - \psi_t^\top \theta$ of the UCB algorithm on the contextual bandit defined by the sequence of action feature sets $\Psi_t$.

**Lemma 3.** *The regret of Algorithm 1 with the expected feature set $\Psi_t$ is bounded at time $T$ with probability at least $1 - \delta$,*

$$\mathcal{R}_T \leq \mathcal{R}_T^{UCB} + 4\sqrt{2T \log\frac{1}{\delta}} \, .$$

*Further, if the algorithm uses the sample based features $\tilde{\Psi}_t$ with $L = t$ at iteration $t$, the regret is bounded at time $T$ with probability at least $1 - \delta$,*

$$\mathcal{R}_T \leq \mathcal{R}_T^{UCB} + 4\sqrt{2T \log\frac{|\mathcal{X}|\pi T}{3\delta}} \, .$$

*Proof.* Consider first the case where we use the expected features $\bar{\psi}_{x_t,\mu_t}$. We add and subtract $(\bar{\psi}_{x_t^*,\mu_t} - \bar{\psi}_{x_t,\mu_t})^\top \theta$ and use $\bar{\psi}_{x_t^*,\mu_t}^\top \theta \leq \psi_t^{*\top}\theta$ to bound the regret by

$$R_T \leq \mathcal{R}_T^{UCB} + \sum_{t=1}^{T} D_t \, ,$$

where we defined $D_t = (\phi_{x_t^*,c_t} - \bar{\psi}_{x_t^*,\mu_t} + \bar{\psi}_{x_t,\mu_t} - \phi_{x_t,c_t})^\top \theta$. It is easy to verify that $\mathbb{E}_{c_t \sim \mu_t}[D_t | \mathcal{F}_{t-1}, \mu_t, x_t] = 0$, that is $D_t$ is a martingale difference sequence with $|D_t| \leq 4$ and $M_T = \sum_{t=1}^T D_t$ is a martingale. The first part of the lemma therefore follows from Azuma-Hoeffding's inequality (Lemma 4, Appendix). For the sample-based version, the reasoning is similar, but we need to ensure that the features $\tilde{\psi}_{x,\mu_t} = \frac{1}{L} \sum_{l=1}^L \phi_{x,\tilde{c}_l}$ are sufficiently concentrated around their expected counterparts $\bar{\psi}_{x,\mu_t}$ for any $x \in \mathcal{X}$ and $t \in \mathbb{N}$. We provide details in Appendix A.1. $\quad\square$

*Proof of Theorem 1.* Clearly, the lemma gives a regret bound for Algorithm 1 if the regret term $\mathcal{R}_T^{UCB}$ is bounded. The main difficulty is that the reward observation $y_t = \phi_{x_t,c_t}^\top \theta + \epsilon_t$ is generated from a different feature vector than the feature $\psi_t \in \Psi_t$ that is chosen by UCB. Note that in general, it is not even true that $\phi_{x_t,c_t} \in \Psi_t$. However, a closer inspection of the reward signal reveals that $y_t$ can be written as

$$ y_t = \psi_t^\top \theta + \xi_t + \epsilon \qquad \text{with} \quad \xi_t := (\phi_{x_t,c_t} - \psi_t)^\top \theta \qquad (2) $$

For the variant that uses the expected features $\psi_t = \bar{\psi}_{x_t,\mu_t}$, our construction already ensures that $\mathbb{E}[\xi_t | \mathcal{F}_{t-1}, \mu_t, x_t] = \mathbb{E}[\phi_{x_t,c_t} - \bar{\psi}_{x_t,\mu_t} | \mathcal{F}_{t-1}, \mu_t, x_t]^\top \theta = 0$. Note that the distribution of $\xi_t$ depends on $x_t$ and is therefore heteroscedastic in general. However, by boundedness of the rewards, $|\xi_t| \leq 2$ and hence $\xi_t$ is 2-subgaussian, which allows us to continue with a homoscedastic noise bound. We see that $y_t$ acts like an observation of $\psi_t^\top \theta$ perturbed by $\sqrt{4 + \sigma^2}$-subgaussian noise (for two independent random variables $X$ and $Y$ that are $\sigma_1$- and $\sigma_2$-subgaussian respectively, $X + Y$ is $\sqrt{\sigma_1^2 + \sigma_2^2}$- subgaussian). Therefore, the construction of the confidence bounds for the least squares estimator w.r.t. $\psi_t$ remains valid at the cost of an increased variance proxy, and we are required to use $\beta_t$ with $\rho = \sqrt{4 + \sigma^2}$ in the definition of the confidence set. The regret bound for the UCB algorithm (Lemma 2) and an application of the union bound completes the proof for this case. When we use the sample-based features $\tilde{\psi}_{x,\mu_t}$, the noise term $\xi_t$ can be biased, because $x_t$ depends on the sampled features and $\mathbb{E}[\tilde{\psi}_{x_t,\mu_t} | \mathcal{F}_{t-1}, \mu_t] \neq \bar{\psi}_{x_t,\mu_t}$. This bias carries on to the least-squares estimator, but can be controlled by a more careful analysis. See Appendix A.2 for details. $\quad\square$

## 4.2 When the context realization is observed

We now turn our attention to the alternative setting, where it is possible to observe the realized context $c_t$ (e.g. actual weather measurements) after the learner has chosen $x_t$. In Algorithm 1, so far our estimate $\hat{\theta}_t$ only uses the data $\{(x_s, \mu_s, y_s)\}_{s=1}^t$, but with the context observation we have $\{(x_s, c_s, y_s)\}_{s=1}^t$ available. It makes sense to use the additional information to improve our estimate $\hat{\theta}_t$, and as we show below this reduces the amount the UCB algorithm explores. The pseudo code of the modified algorithm is given in Algorithm 2 (Appendix B), where the only difference is that we replaced the estimate of $\theta$ by the least squares estimate $\hat{\theta}_t = \arg\min_{\theta' \in \mathbb{R}^d} \sum_{s=1}^t (\phi_{x_s,c_s}^\top \theta' - y_s)^2 + \lambda \|\theta'\|_2^2$. Since now the observation noise $\epsilon_t = y_t - \phi_{x_t,c_t}^\top \theta$ is only $\sigma$- subgaussian (instead of $\sqrt{4 + \sigma^2}$-subgaussian), we can use the smaller scaling factor $\hat{\beta}_t$ with $\rho = \sigma$ to obtain a tighter upper confidence bound.

**Theorem 2.** *The regret of Algorithm 2 based on the expected feature sets $\Psi_t$ and $\beta_t = \beta_t(\sigma, \delta/3)$ is bounded with probability at least $1 - \delta$ by*

$$ \mathcal{R}_T \leq \beta_T \sqrt{8T \log\left(\frac{\det V_T}{\det V_0}\right)} + 4(1 + \lambda^{-1/2}\beta_T)\sqrt{2T \log \frac{3}{\delta}} $$

The importance of this result is that it justifies the use of the smaller scaling $\beta_t$ of the confidence set, which affects the action choice of the UCB algorithm. In practice, $\beta_t$ has a large impact on the amount of exploration, and a tighter choice can significantly reduce the regret as we show in our experiments. We note that in this case, the reduction to the regret bound of UCB is slightly more involved than previously. As before, we use Lemma 3 to reduce a regret bound on $\mathcal{R}_T$ to the regret $\mathcal{R}_T^{UCB}$ that the UCB algorithm obtains on the sequence of context-feature sets $\Psi_t$. Since now, the UCB action is based on tighter confidence bounds, we expect the regret $\mathcal{R}_T^{UCB}$ to be smaller, too. This does not follow directly from the UCB analysis, as there the estimator is based on the features $\bar{\psi}_{x_t,\mu_t}$ instead of $\phi_{x_t,c_t}$. We defer the complete proof to Appendix B.1. There we also show a similar result for the sample based feature sets $\tilde{\Psi}_t$ analogous to Theorem 1.

### 4.3 Kernelized stochastic bandits with context distributions

In the kernelized setting, the reward function $f : \mathcal{X} \times \mathcal{C} \to \mathbb{R}$ is a member of a known reproducing kernel Hilbert space (RKHS) $\mathcal{H}$ with kernel function $k : (\mathcal{X} \times \mathcal{C})^2 \to \mathbb{R}$. In the following, let $\| \cdot \|_{\mathcal{H}}$ be the Hilbert norm and we denote by $k_{x,c} := k(x, c, \cdot, \cdot) \in \mathcal{H}$ the kernel features. For the analysis we further make the standard boundedness assumption $\|f\| \leq 1$ and $\|k_{x,c}\| \leq 1$. We provide details on how to estimate $f$ given data $\{(x_s, \mu_s, y_s)\}_{s=1}^t$ with uncertain context. As in the linear case, the estimator $\hat{f}_t$ can be defined as an empirical risk minimizer with parameter $\lambda > 0$,

$$\hat{f}_t = \arg\min_{f \in \mathcal{H}} \sum_{s=1}^t \left( \mathbb{E}_{c \sim \mu_t}[f(x_s, c)] - y_s \right)^2 + \lambda \|f\|_{\mathcal{H}}^2 . \tag{3}$$

In the literature this is known as *distributional risk minimization* (Muandet et al., 2017, Section 3.7.3). The following representer theorem shows, that the solution can be expressed as a linear combination of kernel mean embeddings $\bar{k}_{x,\mu} := \mathbb{E}_{c \sim \mu}[k_{x,c}] \in \mathcal{H}$.

**Theorem 3** (Muandet et al. (2012, Theorem 1)). *Any $f \in \mathcal{H}$ that minimizes the regularized risk functional* (3) *admits a representation of the form $f = \sum_{s=1}^t \alpha_s \bar{k}_{x_s,\mu_s}$ for some $\alpha_s \in \mathbb{R}$.*

It is easy to verify that the solution to (3) can be written as

$$\hat{f}_t(x, c) = k_t(x, c)^\top (K_t + \lambda \mathbf{I})^{-1} y_t \tag{4}$$

where $k_t(x, c) = [\bar{k}_{x_1,\mu_1}(x, c), \ldots, \bar{k}_{x_t,\mu_t}(x, c)]^\top$, $(K_t)_{a,b} = \mathbb{E}_{c \sim \mu_b}[\bar{k}_{x_a,\mu_a}(x_b, c)]$ for $1 \leq a, b, \leq t$ is the kernel matrix and $y_t = [y_1, \ldots, y_t]^T$ denotes the vector of observations. Likewise, the estimator can be computed from sample based kernel mean embeddings $\tilde{k}_{x,\mu}^L := \frac{1}{L} \sum_{i=1}^L k(x, \tilde{c}_i, \cdot, \cdot) \in \mathcal{H}$ for i.i.d. samples $\tilde{c}_i \sim \mu$. This allows for an efficient implementation also in the kernelized setting, at the usual cost of inverting the kernel matrix. With iterative updates the overall cost amount to $\mathcal{O}(LT^3)$. The cubic scaling in $T$ can be avoided with finite dimensional feature approximations or inducing points methods, e.g. Rahimi and Recht (2008); Mutny and Krause (2018).

The UCB algorithm can be defined using an analogous concentration result for the RKHS setting (Abbasi-Yadkori, 2012). We provide details and the complete kernelized algorithm (Algorithm 3) in Appendix C. The corresponding regret bound is summarized in the following theorem.

**Theorem 4.** *At any time $T \in \mathbb{N}$, the regret of Algorithm 3 with exact kernel mean embeddings $\bar{k}_{x,c}$ and $\beta_t$ as defined in Lemma 6 in Appendix C, is bounded with probability at least $1 - \delta$ by*

$$\mathcal{R}_T \leq \beta_T \sqrt{8T \log(\det(\mathbf{I} + (\lambda\rho)^{-1} K_T))} + 4\sqrt{2T \log \frac{2}{\delta}}$$

Again, the data dependent log-determinant in the regret bound can be replaced with kernel specific bounds, referred to as *maximum information gain* $\gamma_T$ (Srinivas et al., 2010).

## 5 Experiments

We evaluate the proposed method on a synthetic example as well as on two benchmarks that we construct from real-world data. Our focus is on understanding the effect of the sample size $L$ used to define the context set $\tilde{\Psi}_t^l$. We compare three different observational modes, with decreasing amount of information available to the learner. First, in the *exact* setting, we allow the algorithm to observe the context realization before choosing an action, akin to the usual contextual bandit setting. Note that this variant possibly obtains negative reward on the regret objective (1), because $x_t^*$ is computed to maximize the expected reward over the context distribution independent of $c_t$. Second, in the *observed* setting, decisions are based on the context distribution, but the regression is based on the exact context realization. Last, only the context distribution is used for the *hidden* setting. We evaluate the effect of the sample sizes $L = 10, 100$ and compare to the variant that uses that exact expectation of the features. As common practice, we treat the confidence parameter $\beta_T$ as tuning parameter that we choose to minimize the regret after $T = 1000$ steps. Below we provide details on the experimental setup and the evaluation is shown in Figure 1. In all experiments, the 'exact' version significantly outperforms the distributional variants or even achieves negative regret as anticipated. Consistent with our theory, observing the exact context after the action choice improves performance compared to the unobserved variant. The sampled-based algorithm is competitive with the expected features already for $L = 100$ samples.

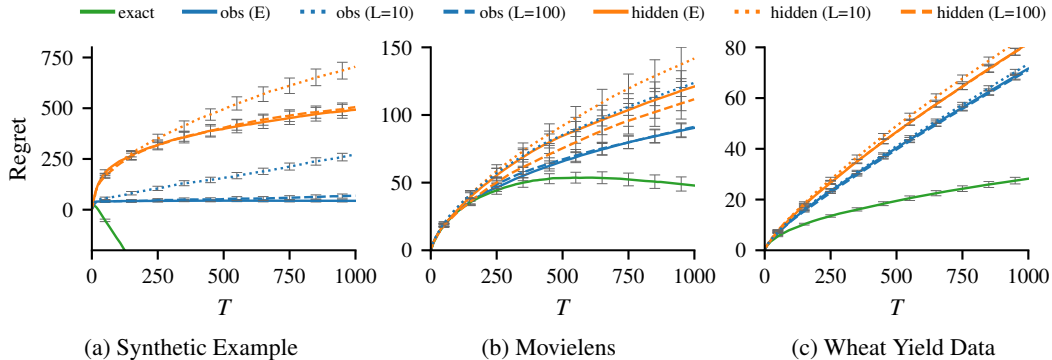

(a) Synthetic Example        (b) Movielens        (c) Wheat Yield Data

Figure 1: The plots show cumulative regret as defined in (1). As expected, the variant that does not observe the context (*hidden*) is out-performed by the variant that uses the context realization for regression (*obs*)4. The sample size $l$ used to construct the feature sets from the context distribution has a significant effect on the regret, where with $l = 100$ performance is already competitive with the policy that uses the exact expectation over features (E). The *exact* baseline, which has access to the context realization before taking the decision, achieves negative regret on the benchmark (a) and (b), as the regret objective (1) compares to the action maximizing the expected reward. The error bars show two times standard error over 100 trials for (a) and (c), and 200 trials for (b). The variance in the movielens experiment is fairly large, likely because our linear model is miss-specified; and at the first glance, it looks like the sample-based version outperforms the expected version in one case. From repeated trials we confirmed, that this is only an effect of the randomness in the results.

**Synthetic Example**    As a simple synthetic benchmark we set the reward function to $f(x, c) = \sum_{i=1}^{5}(x_i - c_i)^2$, where both actions and context are vectors in $\mathbb{R}^5$. We choose this quadratic form to create a setting where the optimal action strongly depends on the context $c_i$. As linear parametrization we choose $\phi(x, c) = (x_1^2, \cdots, x_5^2, c_1^2, \cdots, c_5^2, x_1 c_1, \ldots, x_5 c_5)$. The action set consists of $k = 100$ elements that we sample at the beginning of each trial from a standard Gaussian distribution. For the context distribution, we first sample a random element $m_t \in \mathbb{R}^5$, again from a multivariate normal distribution, and then set $\mu_t = \mathcal{N}(m_t, \mathbf{1})$. Observation noise is Gaussian with standard deviation 0.1.

**Movielens Data**    Using matrix factorization we construct 6-dimensional features for user ratings of movies in the *movielens-1m* dataset (Harper and Konstan, 2016). We use the learned embedding as ground truth to generate the reward which we round to half-integers between 0 and 5 likewise the actual ratings. Therefore our model is miss-specified in this experiment. Besides the movie ratings, the data set provides basic demographic data for each user. In the interactive setting, the context realization is a randomly sampled user from the data. The context distribution is set to the empirical distribution of users in the dataset with the same demographic data. The setup is motivated by a setting where the system interacts with new users, for which we already obtained the basic demographic data, but not yet the exact user's features (that in collaborative filtering are computed from the user's ratings). We provide further details in Appendix D.1.

**Crop Yield Data**    We use a wheat yield dataset that was systematically collected by the Agroscope institute in Switzerland over 15 years on 10 different sites. For each site and year, a 16-dimensional suitability factor based on recorded weather conditions is available. The dataset contains 8849 yield measurements for 198 crops. From this we construct a data set $\mathcal{D} = \{(x_i, w_i, y_i)\}$ where $x_i$ is the identifier of the tested crop, $w_i \in \mathbb{R}^{16+10}$ is a 16 dimensional suitability factor obtained from weather measurements augmented with a 1-hot encoding for each site, and $y_i$ is the normalized crop yield. We fit a bilinear model $y_i \approx w_i^T W V_{x_i}$ to get 5-dimensional features $V_x$ for each variety $x$ and site features $w^\top W$ that take the weather conditions $w$ of the site into account. From this model, we generate the ground-truth reward. Our goal is to provide crop recommendations to maximize yield on a given site with characteristics $w$. Since $w$ is based on weather measurements that are not available ahead of time, we set the context distribution such that each feature of $w$ is perturbed by a Gaussian distribution centered around the true $w$. We set the variance of the perturbation to the empirical variance of the features for the current site over all 15 years. Further details are in Appendix D.2.

# 6 Related Work

There is a large array of work on bandit algorithms, for a survey see Bubeck and Cesa-Bianchi (2012) or the book by Lattimore and Szepesvári (2018). Of interest to us is the *stochastic contextual bandit problem*, where the learner chooses actions after seeing a context; and the goal is to compete with a class of policies, that map contexts to actions. This is akin to reinforcement learning (Sutton and Barto, 2018), but the contextual bandit problem is different in that the sequence of contexts is typically allowed to be *arbitrary* (even adversarially chosen), and does not necessarily follow a specific transition model. The contextual bandit problem in this formulation dates back to at least Abe and Long (1999) and Langford and Zhang (2007). The perhaps best understood instance of this model is the *linear contextual bandit*, where the reward function is a linear map of feature vectors (Auer et al., 2002). One of the most popular algorithms is the Upper Confidence Bound (UCB) algorithm, first introduced by Auer (2002) for the multi-armed bandit problem, and later extended to the linear case by Li et al. (2010). Analysis of this algorithm was improved by Dani et al. (2008), Abbasi-Yadkori et al. (2011) and Li et al. (2019), where the main technical challenge is to construct tight confidence sets for an online version of the least squares estimator. Alternative exploration strategies have been considered as well, for instance Thompson sampling (Thompson, 1933), which was analyzed for the linear model by Agrawal and Goyal (2013) and Abeille and Lazaric (2017). Other notable approaches include an algorithm that uses a perturbed data history as exploration mechanism (Kveton et al., 2019), or a *mostly greedy* algorithm that leverages the randomness in the context to obtain sufficient exploration (Bastani et al., 2017). In *kernelized bandits* the reward function is contained given reproducing kernel Hilbert space (RKHS). This setting is closely related to Bayesian optimization (Mockus, 1982). Again, the analysis hinges on the construction of confidence sets and bounding a quantity referred to as *information gain* by the decay of the kernel's eigenspectrum. An analysis of the UCB algorithm for this setting was provided by Srinivas et al. (2010). It was later refined by Abbasi-Yadkori (2012); Valko et al. (2013); Chowdhury and Gopalan (2017); Durand et al. (2018) and extended to the contextual setting by Krause and Ong (2011). Interestingly, in our reduction the noise distribution depends on the action, also referred to as *heteroscedastic* bandits. Heteroscedastic bandits where previously considered by Hsieh et al. (2019) and Kirschner and Krause (2018). Stochastic uncertainty on the action choice has been studied by Oliveira et al. (2019) in the context of Bayesian optimization. Closer related is the work by Yun et al. (2017), who introduce a linear contextual bandit model where the observed feature is perturbed by noise and the objective is to compete with the best policy that has access to the unperturbed feature vector. The main difference to our setting is that we assume that the environment provides a distribution of feature vectors (instead of a single, perturbed vector) and we compute the best action as a function of the distribution. As a consequence, we are able to obtain $\mathcal{O}(\sqrt{T})$ regret bounds without further assumptions on the context distribution, while Yun et al. (2017) get $\mathcal{O}(T^{7/8})$ with identical noise on each feature, and $\mathcal{O}(T^{2/3})$ for Gaussian feature distributions. Most closely related is the work by Lamprier et al. (2018) on linear bandits with stochastic context. The main difference to our setting is that the context distribution in Lamprier et al. (2018) is fixed over time, which allows to built aggregated estimates of the mean feature vector over time. Our setting is more general in that it allows an arbitrary sequence of distributions as well as correlation between the feature distributions of different actions. Moreover, in contrast to previous work, we discuss the kernelized-setting and the setting variant, where the context is observed exactly after the action choice. Finally, also adversarial contextual bandit algorithms apply in our setting, for example the EXP4 algorithm of Auer et al. (2002) or ILTCB of Agarwal et al. (2014). Here, the objective is to compete with the best policy in a given class of policies, which in our setting would require to work with a covering of the set of distributions $\mathcal{P}(\mathcal{C})$. However, these algorithms do not exploit the linear reward assumption and, therefore, are arguably less practical in our setting.

# 7 Conclusion

We introduced *context distributions* for stochastic bandits, a model that is naturally motivated in many applications and allows to capture the learner's uncertainty in the context realization. The method we propose is based on the UCB algorithm, and in fact, both our model and algorithm strictly generalize the standard setting in the sense that we recover the usual model and the UCB algorithm if the environment chooses only Dirac delta distributions. The most practical variant of the proposed algorithm requires only sample access to the context distributions and satisfies a high-probability regret bound that is order optimal in the feature dimension and the horizon up to logarithmic factors.

**Acknowledgments**

The authors thank Agroscope for providing the crop yield data set, in particular Didier Pellet, Lilia Levy and Juan Herrera, who collected the winter wheat data, and Annelie Holzkämper, who developed the environmental suitability factors model. Further, the authors acknowledge the work by Mariyana Koleva and Dejan Mirčić, who performed the initial data cleaning and exploration as part of their Master's theses.

This research was supported by SNSF grant 200020 159557 and has received funding from the European Research Council (ERC) under the European Union's Horizon 2020 research and innovation programme grant agreement No 815943.

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
