[Supplementary Material]

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

# A  Proof Details

**Lemma 4** (Azuma-Hoeffdings). *Let $M_t$ be a martingale on a filtration $\mathcal{F}_t$ with almost surely bounded increments $|M_t - M_{t-1}| < B$. Then*

$$\mathbb{P}[M_T - M_0 > s] \leq \exp\left(-\frac{s^2}{2TB^2}\right)$$

## A.1  Proof of Lemma 3

To bound the regret of Algorithm 1 with sampled features $\tilde{\psi}_{x,\mu_t}$, we add and subtract $(\tilde{\psi}_{x_t^*,\mu_t} + \bar{\psi}_{x_t,\mu_t} + \tilde{\psi}_{x_t,\mu_t})^\top \theta$ to the regret to find

$$\mathcal{R}_T = \mathcal{R}_T^{UCB} + \sum_{t=1}^{T}(\phi_{x_t^*,c_t} - \tilde{\psi}_{x_t^*,\mu_t} + \bar{\psi}_{x_t,\mu_t} - \phi_{x_t,c_t})^\top \theta + \sum_{t=1}^{T}(\tilde{\psi}_{x_t,\mu_t} - \bar{\psi}_{x_t,\mu_t})^\top \theta$$

$$\leq \mathcal{R}_T^{UCB} + 4\sqrt{2T \log\frac{1}{\delta}} + \sum_{t=1}^{T}(\tilde{\psi}_{x_t,\mu_t} - \bar{\psi}_{x_t,\mu_t})^\top \theta .$$

By the same reasoning as for the expected features, we bounded

$$\sum_{t=1}^{T}(\phi_{x_t^*,c_t} - \tilde{\psi}_{x_t^*,\mu_t} + \bar{\psi}_{x_t,\mu_t} - \phi_{x_t,c_t})^\top \theta \leq 4\sqrt{2T \log\frac{1}{\delta}} \tag{5}$$

using Azuma-Hoeffding's inequality. We are left with a sum $\sum_{t=1}^{T}(\tilde{\psi}_{x_t,\mu_t} - \bar{\psi}_{x_t,\mu_t})^\top \theta$ that is more intricate to bound because $x_t$ depends on the samples $\tilde{c}_{t,l}$ that define $\tilde{\psi}_{x_t,\mu_t}$. We exploit that for large $L$, $\tilde{\psi}_{x_t,\mu_t}^L - \bar{\psi}_{x_t,\mu_t} \to 0$. First consider a fixed $x \in \mathcal{X}$. Then, by $(\tilde{\psi}_{x,\mu_t} - \bar{\psi}_{x,\mu_t})^\top \theta \leq 2$ and Azuma-Hoeffding's inequality with probability at least $1 - \delta$,

$$(\tilde{\psi}_{x,\mu_t} - \bar{\psi}_{x,\mu_t})^\top \theta \leq \sqrt{\frac{8}{L} \log\frac{1}{\delta}} . \tag{6}$$

We enforce this to hold for any $x \in \mathcal{X}$ and any time $t \in \mathbb{N}$ by replacing $\delta$ by $\frac{6\delta}{|\mathcal{X}|\pi^2 t^2}$ and taking the union bound over the event where (6) holds. By our choice $L = t$ and $\sum_{t=1}^{T}\frac{1}{\sqrt{t}} \leq 2\sqrt{T}$, we have with probability at least $1 - \delta$, for any $T \in \mathbb{N}$,

$$\sum_{t=1}^{T}(\tilde{\psi}_{x_t,\mu_t} - \bar{\psi}_{x_t,\mu_t})^\top \theta \leq 4\sqrt{T \log\frac{|\mathcal{X}|\pi^2 T^2}{6\delta}} . \tag{7}$$

A final application of the union bound over the events such that (5) and (7) simultaneously hold, gives

$$R_T \leq \mathcal{R}_T^{UCB} + 4\sqrt{2T \log\frac{|\mathcal{X}|\pi T}{3\delta}} .$$

## A.2  Proof of Theorem 1

To bound the regret term $\mathcal{R}_T^{UCB}$ for the case where we uses sample-based feature vectors, the main task is to show a high-probability bound on $\|\theta - \hat{\theta}_t\|_{V_t}$ with observations $y_t = \tilde{\psi}_{x,t}^\top \theta + \xi_t + \epsilon_t$. Recall that $\xi_t = (\phi_{x,c_t} - \tilde{\psi}_{x_t,\mu_t})^\top \theta$, but now we have in general $\mathbb{E}[\xi_t|\mathcal{F}_{t-1}, \mu_t, x_t] \neq 0$, because $x_t$ depends on the sampled features $\tilde{\psi}_{x,\mu_t}$. The following lemma bounds the estimation error of the least-square estimator in the case where the noise term contains an (uncontrolled) biased term.

**Lemma 5.** *Let $\hat{\theta}_t$ be the least squares estimator $\hat{\theta}_t$ defined for any sequence $\{(\phi_t, y_t)\}_t$ with observations $y_t = \phi_t^\top \theta + b_t + \epsilon_t$, where $\epsilon_t$ is $\rho$-subgaussian noise and $b_t$ is an arbitrary bias. The following bound holds with probability at least $1 - \delta$, at any time $t \in \mathbb{N}$,*

$$\|\theta - \hat{\theta}_t\|_{V_t} \leq \beta_t + \sqrt{\sum_{t=1}^{T} b_t^2} \qquad \text{where } \beta_t = \beta_t(\rho, \delta) = \rho\sqrt{2\log\left(\frac{\det(V_t)^{1/2}}{\delta \det(V_0)^{1/2}}\right)} + \lambda^{1/2}\|\theta\|_2 .$$

*Proof of Lemma.* Basic linear algebra and the triangle inequality show that

$$\|\theta - \hat{\theta}_t\|_{V_t} \leq \|\sum_{s=1}^{t} \phi_s \epsilon_s\|_{V_s^{-1}} + \|\sum_{s=1}^{t} \phi_s b_s\|_{V_s^{-1}} + \lambda^{1/2}\|\theta\|_2 \tag{8}$$

Recall that we assume $\|\theta\|_2 \leq 1$ in order to bound the last term. The noise process can be controlled with standard results (Abbasi-Yadkori et al., 2011, Theorem 1), specifically $\|\sum_{s=1}^{t} \phi_s \epsilon_s\|_{V_s^{-1}} \leq \beta_t$. Finally, to bound the sum over the biases, set $b = [b_1, \ldots, b_t]$ and $A = [\phi_1, \ldots, \phi_t]^\top \in \mathbb{R}^{d \times t}$. The matrix inequality

$$A^\top (AA^\top + \lambda \mathbf{I}_d)^{-1} A \leq \mathbf{I}_t \tag{9}$$

follows by using a SVD decomposition. This implies

$$\|\sum_{s=1}^{t} \phi_s b_s\|_{V_s^{-1}}^2 = \|Ab\|_{(AA^\top + \lambda \mathbf{I}_d)^{-1}}^2 \leq \|b\|_2^2$$

Applying the individual bounds to (8) completes the proof of the lemma. $\qquad\square$

We continue the proof of the theorem with the intuition to use the lemma with $b_t = (\bar{\psi}_{x,\mu_t} - \tilde{\psi}_{x_t,\mu_t})^\top \theta$. To control the sum over bias terms $b_t$, note that by our choice $L = t$, similar to (7), with probability at least $1 - \delta$, for all $t \in \mathbb{N}$ and $x \in \mathcal{X}$,

$$|b_t| \leq \sqrt{\frac{8}{t} \log \frac{\pi^2 t^2 |\mathcal{X}|}{6\delta}} \ .$$

Hence with $\sum_{s=1}^{t} \frac{1}{s} \leq \log(t)$, we get

$$\sum_{t=1}^{T} b_t^2 \leq 8 \log(T) \log\left(\frac{\pi^2 T^2 |\mathcal{X}|}{6\delta}\right) \ .$$

As before, the remaining terms are zero-mean, $\mathbb{E}[(\phi_{x_t,c_t} - \bar{\psi}_{x_t,\mu_t})^\top \theta + \epsilon_t | \mathcal{F}_{t-1}, \mu_t, x_t] = 0$, and $\sqrt{4 + \sigma^2}$-subgaussian. Hence, with the previous lemma and another application of the union bound we get with probability at least $1 - \delta$,

$$\|\theta - \hat{\theta}_t\|_{V_t} \leq \beta_t + \sqrt{\sum_{t=1}^{T} b_t^2} + \lambda\|\theta\|_2$$

$$\leq \sqrt{2(4 + \sigma^2) \log\left(\frac{2 \det(V_t)^{1/2}}{\delta \det(V_0)^{1/2}}\right)} + \sqrt{8 \log(T) \log\left(\frac{\pi^2 T^2 |\mathcal{X}|}{3\delta}\right)} + \lambda \ . \tag{10}$$

Finally, we invoke Lemma 2 with $\beta_t = \tilde{\beta}_t$, where

$$\tilde{\beta}_t := \sqrt{2(4 + \sigma^2) \log\left(\frac{2 \det(V_t)^{1/2}}{\delta \det(V_0)^{1/2}}\right)} + \sqrt{8 \log(T) \log\left(\frac{\pi^2 T^2 |\mathcal{X}|}{3\delta}\right)} + \lambda \tag{11}$$

to obtain a bound on $\mathcal{R}_T^{UCB}$,

$$\mathcal{R}_T^{UCB} \leq \tilde{\beta}_T \sqrt{8T \log\left(\frac{\det V_T}{\det V_0}\right)} \ . \tag{12}$$

This concludes the proof.

# B  UCB with Context Distributions and Observed Context

---

**Algorithm 2** UCB for linear stochastic bandits with context distributions and observed context

---

Initialize $\hat{\theta} = 0 \in \mathbb{R}^d$, $V_0 = \lambda\mathbf{I} \in \mathbb{R}^{d \times d}$
**For** step $t = 1, 2, \ldots, T$:
    *Environment* chooses $\mu_t \in \mathcal{P}(\mathcal{C})$                                     // *context distribution*
    *Learner* observes $\mu_t$

    Set $\Psi_t = \{\psi_{x,\mu_t} : x \in \mathcal{X}\}$ with $\bar{\psi}_{x,\mu_t} = \mathbb{E}_{\mu_t}[\phi_{x,c}]$                // *expected version*
    Alternatively, sample $c_{t,1}, \ldots, c_{t,L}$ for $L = t$,             // *or sampled version*
    Set $\Psi_t = \{\tilde{\psi}_{x,\mu_t} : x \in \mathcal{X}\}$ with $\tilde{\psi}_{x,\mu_t} := \frac{1}{L}\sum_{i=1}^L \phi_{x,\tilde{c}_i}$

    Run UCB step with $\Psi_t$ as context set                               // *reduction*
    Choose action $x_t = \arg\max_{x \in \mathcal{X}} \psi_{x,\mu_t}^\top \hat{\theta}_{t-1} + \beta_t(\sigma)\|\psi_{x,\mu_t}\|_{V_{t-1}^{-1}}$     // *UCB action*

    *Environment* samples $c_t \sim \mu_t$
    *Learner* observes $y_t = \phi_{x_t,c_t}^\top \theta + \epsilon_t$ and $c_t$         // *reward and context observation*
    Update $V_t = V_{t-1} + \phi_{x_t,c_t}\phi_{x_t,c_t}^\top$, $\hat{\theta}_t = V_t^{-1}\sum_{s=1}^t \phi_{x_s,c_s}y_s$     // *least-squares update*

---

## B.1  Proof of Theorem 2

From Lemma 3 we obtain with probability at least $1 - \delta$,

$$\mathcal{R}_T \leq \sum_{t=1}^T \psi_t^{*\top}\theta - \bar{\psi}_{x_t,\mu_t}^\top \theta + 4\sqrt{2T\log\frac{1}{\delta}} \tag{13}$$

What is different now, is that the sum $\sum_{t=1}^T \psi_t^{*\top}\theta - \bar{\psi}_{x_t,\mu_t}^\top \theta$ contains actions $x_t$, that are computed from a more precise estimator $\hat{\theta}_t$, hence we expect the regret to be smaller. This does not follow directly from UCB analysis, as there the estimator is computed with the features $\bar{\psi}_{x_t,\mu_t}$.

We start with the usual regret analysis, making use of the confidence bounds (Lemma 1) and the definition of the UCB action. Denote $\psi_t := \bar{\psi}_{x_t,\mu_t}$ in the following.

$$\sum_{t=1}^T \psi_t^{*\top}\theta - \psi_t^\top \theta \leq \sum_{t=1}^T \psi_t^{*\top}\hat{\theta}_t + \beta_t\|\psi_t^*\|_{V_t^{-1}} - (\phi_t^\top \hat{\theta}_t - \beta_t\|\psi_t\|_{V_t^{-1}}) \leq 2\beta_T \sum_{t=1}^T \|\psi_t\|_{V_t^{-1}} \tag{14}$$

From here, the standard analysis proceeds by using Cauchy-Schwarz to obtain an upper bound on $\sum_{t=1}^T \|\psi_t\|_{V_t^{-1}}$ and then the argument proceeds by simplifying the sum $\sum_{t=1}^T \|\psi_t\|_{V_t^{-1}}$. The simplification doesn't work here because $V_t$ is defined on the realized features $\phi_{x_t,c_t}$ and not $\psi_t$. Instead we require the following intermezzo.

$$\sum_{t=1}^T \|\psi_t\|_{V_t^{-1}} = \sum_{t=1}^T \|\phi_{x_s,c_s}\|_{V_t^{-1}} + \|\psi_t\|_{V_t^{-1}} - \|\phi_{x_s,c_s}\|_{V_t^{-1}} \leq \sum_{t=1}^T \|\phi_{x_s,c_s}\|_{V_t^{-1}} + \sum_{t=1}^T S_t \, , \tag{15}$$

where we defined $S_t = \|\psi_t\|_{V_t^{-1}} - \|\phi_{x_s,c_s}\|_{V_t^{-1}}$. We show that $\sum_{t=1}^T S_t$ is a supermartingale. For the expected features $\bar{\psi}_{x,\mu_t} = \mathbb{E}_{c \sim \mu_t}[\phi_{x,c}]$, note that Jensen's inequality yields $\|\mathbb{E}_{c \sim \mu_t}[\phi_{x,c}]\|_{V_t^{-1}} \leq \mathbb{E}_{c \sim \mu_t}[\|\phi_{x,c}\|_{V_t^{-1}}]$ for all $x \in \mathcal{X}$. From this we obtain $\mathbb{E}[S_t|\mathcal{F}_{t-1}, \mu_t] \leq 0$. Finally, note that $\|\phi_{x_s,c_s}\|_{V_t^{-1}} \leq \lambda^{-1/2}\|\phi_{x_s,c_s}\|_2 \leq \lambda^{-1/2}$ and $|S_t| \leq 2\lambda^{-1/2}$, hence by Azuma-Hoeffdings inequality with probability at least $1 - \delta$, $\sum_{t=1}^T S_t \leq 2\lambda^{-1/2}\sqrt{2T\log\frac{1}{\delta}}$.

From here we complete the regret analysis by bounding $\sum_{t=1}^T \|\phi_{x_s,c_s}\|_{V_t^{-1}}$ with the standard argument. Write $\phi_t := \phi_{x_t,c_t}$. First using Cauchy-Schwarz and then $\|\phi_t\|_2 \leq 1$ as well as $u \leq 2\log(1+u)$

for $u \le 1$, it follows that

$$\sum_{t=1}^{T} \|\phi_t\|_{V_t^{-1}} \le \sqrt{T \sum_{t=1}^{T} \|\phi_t\|_{V_t^{-1}}^2} \le \sqrt{2T \sum_{t=1}^{T} \log(1 + \|\phi_t\|_{V_t^{-1}}^2)} = \sqrt{2T \log\left(\frac{\det V_T}{\det V_0}\right)} \quad (16)$$

The last equality essentially follows from an application of the Sherman-Morrison formula on the matrix $V_t = \sum_{s=1}^{t} \phi_{x_t,c_t} \phi_{x_t,c_t}^\top + \lambda \mathbf{I}_d$, compare e.g. (Abbasi-Yadkori et al., 2011, Lemma 11). It remains to assemble the results from equations (14)-(16), and a final application of the union bound completes the proof.

With a bit of extra work, one can obtain a similar bound for the sample-based features. Using Jensen's inequality we get $\|\frac{1}{L} \sum_{i=1}^{L} \phi_{x,\tilde{c}_i}\|_{V_t^{-1}} \le \frac{1}{L} \sum_{i=1}^{L} \|\phi_{x,\tilde{c}_i}\|_{V_t^{-1}}$, but now the action $x_t$ depends on the samples $\tilde{c}_{t,i}$ that define the features $\tilde{\psi}_{x_t,\mu_t}$ and the previous direct argument does not work. The strategy around is the sames as in the proof of Lemma 3. For fixed $x \in \mathcal{X}$, $\frac{1}{L} \sum_{i=1}^{L} \|\phi_{x,\tilde{c}_i}\|_{V_t^{-1}}$ concentrates around $\mathbb{E}_{c \sim \mu_t}[\|\phi_{x,c}\|_{V_t^{-1}}]$ at a rate $1/\sqrt{L}$, fast enough to get $\mathcal{O}(\sqrt{T})$ regret if we set $L = t$ in iteration $t$. A careful application of the union bound (again over all $x \in \mathcal{X}$) completes the proof.

## C  Kernelized UCB with Context Distributions

### C.1  Proof of Theorem 4

To bound the regret we proceed like in the linear case. First, define $\bar{f}_t(x) = \mathbb{E}_{\mu_t}[f(x,c)|\mathcal{F}_{t-1}]$. Analogously to Lemma 3, an application of Azuma-Hoeffding's inequality yields with probability at least $1 - \delta$.

$$\mathcal{R}_T \le \sum_{t=1}^{T} \bar{f}_t(x_t^*) - \bar{f}_t(x_t^*) + 4\sqrt{2T \log \frac{1}{\delta}} \quad (17)$$

To bound the sum, we need to understand the concentration behavior of $|\hat{f}_t(x) - \bar{f}_t(x_t^*)|$, where we denote $\hat{f}_t(x) = \mathbb{E}_{\mu_t}[\hat{f}_t(x,c)|\mathcal{F}_{t-1}, \mu_t]$. Recall that

$$\hat{f}_t(x,c) = k_t(x,c)^\top (K_t + \lambda \mathbf{I})^{-1} y_t \quad (18)$$

where $k_t(x,c) = [\bar{k}_{x_1,\mu_1}(x,c), \dots, \bar{k}_{x_t,\mu_t}(x,c)]^\top$, $(K_t)_{i,j} = \mathbb{E}_{c' \sim \mu_j}[\bar{k}_{x_i,\mu_i}(x_j,c')]$ for $1 \le i,j, \le t$ is the kernel matrix and $y_t = [y_1, \dots, y_t]^T$ denotes the vector of observations. Define further $\bar{k}_t(x) = \mathbb{E}_{c \sim \mu_t}[k_t(x,c)]$.

Note that we can compute $\hat{f}_t(x) = \langle \hat{f}, k_{x,\mu_t} \rangle = \bar{k}_t(x)^\top (K_t + \lambda \mathbf{I})^{-1} y_t$ according to the inner product $\langle \cdot, \cdot \rangle$ on $\mathcal{H}$. Concentration bounds for the kernel least squares estimator, that hold for adaptively collected data, are well understood by now, see Srinivas et al. (2010); Abbasi-Yadkori (2012); Chowdhury and Gopalan (2017); Durand et al. (2018). For instance, as a direct corollary of (Abbasi-Yadkori, 2012, Theorem 3.11), we obtain

**Lemma 6.** *For any stochastic sequence $\{(x_t, \mu_t, y_t)\}_{t \in \mathbb{N}}$, where $y_t = f(x_t, c_t) + \epsilon_t$ with $\sigma$-subgaussian noise $\epsilon_t$ and $c_t \sim \mu_t$, the kernel-least squares estimate (3) satisfies with probability at least $1 - \delta$, at any time $t$ and for any $x \in \mathcal{X}$,*

$$|\hat{f}_t(x) - \bar{f}_t(x_t^*)| \le \beta_t \sigma_t(x).$$

*Here we denote,*

$$\beta_t = \rho \left( \sqrt{2 \log \left( \frac{\det(\mathbf{I} + (\lambda\rho)^{-1} K_t)^{1/2}}{\delta} \right)} + \lambda^{1/2} \|f\|_{\mathcal{H}} \right),$$

$$\sigma_t^2(x) = \frac{1}{\lambda} \left( \langle k_{x,\mu_t}, k_{x,\mu_t} \rangle - k_t(x)^\top (K_t + \lambda \mathbf{I})^{-1} k_t(x) \right),$$

*and $\rho = \sqrt{4 + \sigma^2}$ is the subgaussian variance proxy of the observation noise $\rho_t = y_t - \bar{f}_t(x)$.*

Using the confidence bounds, we define the UCB action,

$$x_t = \arg\max_{x \in \mathcal{X}} \hat{f}_t(x) + \beta_t \sigma_t(x) \,. \tag{19}$$

It remains to bound the regret of the UCB algorithm. The proof is standard except that we use Lemma 6 to show concentration of the estimator. For details see (Abbasi-Yadkori, 2012, Theorem 4.1) or (Chowdhury and Gopalan, 2017, Theorem 3).

---

**Algorithm 3** UCB for RKHS bandits with context distributions

---

Initialize $\hat{f}_0, \sigma_0$
**For** step $t = 1, 2, \ldots, T$:
  *Environment* chooses $\mu_t \in \mathcal{P}(\mathcal{C})$                                        *// context distribution*
  *Learner* observes $\mu_t$

  *// definitions for expected version*
  $[k_t(x)]_s := \mathbb{E}_{c \sim \mu_t}[k_{x_s, \mu_s}(x, c)]$     for $s = 1, \ldots, t-1$
  $s_t(x) := \mathbb{E}_{c \sim \mu_t, c' \sim \mu_t}[k(x, c, x, c')]$

  *// definitions for sampled version*
  Sample $\tilde{c}_{t,1}, \ldots, \tilde{c}_{t,L} \sim \mu_t$                                        *// sample context distribution*
  $[k_t(x)]_s := \frac{1}{L} \sum_{i=1}^{L} k_{x_s, \mu_s}(x, \tilde{c}_i)$     for $s = 1, \ldots, t-1$
  $s_t(x) := \frac{1}{L^2} \sum_{i,j=1}^{L} k(x, \tilde{c}_{t,i}, x, \tilde{c}_{t,j})$

  $\hat{f}_t(x) := k_t(x)^\top (K_t + \lambda \mathbf{I})^{-1} y_t$                                        *// compute estimate*
  $\sigma_t^2(x) := \frac{1}{\lambda}\big(s_t(x) - k_t(x)^\top (K_t + \lambda \mathbf{I})^{-1} k_t(x)\big)$                                        *// confidence width*

  Set $\beta_t$ as in Lemma 6
  Choose action $x_t \in \arg\max_{x \in \mathcal{X}} \hat{f}_{t-1}(x) + \beta_t \sigma_{t-1}(x)$                                        *// UCB action*

  *Environment* provides $y_t = f(x_t, c_t) + \epsilon$ where $c_t \sim \mu_t$                                        *// reward observation*
  Store $y_{t+1} := [y_1, \ldots, y_t]$                                        *// observation vector*

  *// Kernel matrix update, expected version*
  $K_{t+1}(x) := [\langle k_{x_a, \mu_a}, k_{x_b, \mu_b} \rangle]_{1 \le a, b \le t}$ with $\langle k_{x_a, \mu_a}, k_{x_b, \mu_b} \rangle = \mathbb{E}_{\mu_a, \mu_b}[k(x_a, c_a, x_b, c_b)]$
  $k_{x_t, \mu_t} := \mathbb{E}_{c \sim \mu_t}[k_{x_t, c}]$                                        *// kernel mean embeddings*

  *// Kernel matrix update, sampled version*
  $K_{t+1}(x) := [\langle k_{x_a, \mu_a}, k_{x_b, \mu_b} \rangle]_{1 \le a, b \le t}$ with $\langle k_{x_a, \mu_a}, k_{x_a, \mu_a} \rangle = \frac{1}{L^2} \sum_{i,j=1}^{L} k(x_a, \tilde{c}_{a,i}, x_b, \tilde{c}_{b,j})$
  $k_{x_t, \mu_t} := \frac{1}{L} \sum_{i=1}^{L} k_{x_t, \tilde{c}_{t,i}} = \frac{1}{L} \sum_{i=1}^{L} k(x_t, \tilde{c}_{t,i}, \cdot, \cdot)$                                        *// sample kernel mean embeddings*

---

# D   Details on the Experiments

We tune $\beta_t$ over the values $\{0.5, 1, 2, 5, 10\}$. In the synthetic experiment we set $\beta_t = 2$ for the *exact* and *observed* variant and $\beta_t = 10$ for the *hidden* experiment. In the both experiments based on real-world data, we set $\beta_t = 1$ for all variants.

## D.1   Movielens

We use matrix factorization based on singular value decomposition (SVD) (Koren et al., 2009) which is implemented in the Surprise library (Hug, 2017) to learn 6 dimensional features $v_u$, $w_m$ for users $u$ and movies $m$ (this model obtains a RMSE $\approx 0.88$ over a 5-fold cross-validation). In the linear parameterization this corresponds to 36-dimensional features $\phi_{m,u} = v_u w_m^T$. We use the demographic data (gender, age and occupation) to group the users. The realized context is set to a random user in the data set, and the context distribution is defined as the empirical distribution over users within the same group as the chosen user.

## D.2 Crops Yield Dataset

Recall that the data set $\mathcal{D} = \{(x_i, w_i, y_i) : i = 1, \ldots, 8849\}$ consists of crop identifiers $x_i$, normalized yield measurements $y_i$ and site-year features $w_i \in \mathbb{R}^{16+10}$ that are based on 16 suitability factors computed from weather measurements and a 1-hot encoding for each site (out of 10). The suitability factors are based on the work of Holzkämper et al. (2013). To obtain a model for the crop yield responses, we train a bilinear model on the following loss (Koren et al., 2009),

$$\mathcal{L}(W, V) = \sum_{i=1}^{n} (y_i - w_i^\top W V_{x_i})^2 + \|v_{j_i}\|^2 + \|w_{x_i}^\top W\|_2^2 . \tag{20}$$

where $V = (V_{x_j})_{j=1}^{k}$ are the crop features $V_{x_j} \in \mathbb{R}^5$ and $W \in \mathbb{R}^{26 \times 6}$ is used to compute site features $w_i^\top W$ given the suitability features $w_i$ from the dataset.

We also use an empirical noise function created from the data. Since the data set contains up to three measurements of the same crop on a specific site and a year, we randomly pick a measurement and use its residual w.r.t. the mean of all measurements of the same crop under exactly the same conditions as noise. This way we ensure that the observation noise of our simulated environment is of the same magnitude as the noise on the actual measurements.