[Reviews · NeurIPS 2019]

Reviewer 1



The paper studies a linear stochastic bandit problem where the learner observes only a distribution over the context vectors and the true context vector is a sample from this distribution. Authors propose a UCB-based algorithm and show that the uncertainty in the context vector leads to only a small additive regret. They also consider a variant of the problem where the realized context is observed after the learner takes an action. A kernelized version of the problem is also studied, and finally experimental results are reported. The setting of the current paper is very similar to the setting of the following paper: “Profile-Based Bandit with Unknown Profiles” by Sylvain Lamprier, Thibault Gisselbrecht, Patrick Gallinari; JMLR 19(53):1-40, 2018. The above paper studies a less restrictive version of the problem where only a sample from the context distribution is observed. I am concerned about the significance of the current paper in light of this earlier result. Please discuss the above paper in your related work section and explain the differences and similarities with their setting. Minor comment: The paragraph after equation (1): authors discuss two different notions of regret. They argue that one notion of regret is too difficult and leads to \Omega(T) regret. But in that example, the other notion of regret leads to 0 regret and the problem becomes trivial. You need a better example here.

Reviewer 2



- Originality: There are two main contributions of this paper. The first one is to consider a CB setting where the context is not known exactly, but only in a distributional sense. The second one is that the author gives an effective algorithm under this setting, analyzing the regret theoretically, and empirically examine the performance on three datasets. - Significance: This paper seems to be a useful contribution to the CB literature, and will motivate some new algorithms in this new setting. Also, the author empirically examines how many data will be suitable to form a context set. - Quality: The theorem is clearly-stated. And I am a little bit concerned about the weak baseline. Even the paper justifies that if we compare with the original baseline, we will get an order T regret without knowing context realization. I am wondering is it possible to give a stronger regret analysis (compared with the original strong baseline) under some specific cases (say with some restriction on the context distribution and reward structure)? For the experiments, in the Movielens case, it is interesting to see the hidden (with sample 100) is better than hidden (expected), any justifications? Also for the Crop Data case, it is better to include some justifications about the very small advantage of using exact context. -Clarity: The paper is well-written and it will be great if there are more explanation of the results from the experiments.

Reviewer 3



This paper is interesting and well written in general. To the best of my knowledge, this is the first paper that studies contextual (stochastic) bandits with context distributions, which is an interesting problem. After a high-level check, I think the analysis is (high-level) technically correct. The experiment results are also interesting. My only major concern is that this paper might be relatively technically straightforward based on existing literature. In particular, the proof for Theorem 1 (the major theoretical result) seems to be quite straightforward by identifying some Martingale difference sequences. Please clarify! ----------------------------------------------------------------------------------- I have read the rebuttal and prefer to keep my score.

[Author Response · NeurIPS 2019]

# Author Response: Stochastic Bandits with Context Distributions

First, we would like to thank all reviewers for their valuable feedback. We address all concerns raised below.

**Reviewer #1**

Thank you for pointing out the reference (Lamprier et al., 2018). We agree that the setting shares some similarity with ours, but also highlight some key differences. The setting of Lamprier et al. (2018) uses a fixed context distribution per action (profiles in their terminology) and only one context sample per round is observed. This allows/requires to build aggregated estimates of the mean context over time. In our setup, the algorithm is granted full access to the context distribution (either exactly or via samples), and the distribution can change from round to round (chosen by an adversary). Our setting specializes to the setting of Lamprier et al. (2018) if the context distribution is fixed and the feature distribution factors over the actions (and only a single sample is observed per round). However, our setting is much more general in that it allows an arbitrary sequence of distributions as well as correlation between the feature distributions of different actions. On a technical level, Lamprier et al. (2018, Proposition 5) controls the deviation in the feature estimates in each dimension separately (which in our opinion requires a union bound over $\mathcal{X}$ that leads to a deviation of $d\log(|\mathcal{X}|)$), whereas our analysis directly bounds the predictive errors and scales with $\log(|\mathcal{X}|)$ (e.g. eq (7)). Finally, the kernelized setting was not considered in the previous work, and we reveal an interesting connection to kernel mean embeddings. We will add discussion of this related work to the updated version of our paper.

Regarding the example that illustrates the different notions of regret, a slightly more interesting case can be obtained by choosing the context as a biased coin flip. This leads to non-trivial regret for some algorithms. More complicated, lower-bound like constructions are possible, too; but this was not our intention at that point.

**Reviewer #2**

Regarding the stronger baseline, even if the true parameter is known to the algorithm, in general it is not possible to compete with a baseline that exploits the exact feature realization (as shown by our example). For this case, constant-per-round expected regret can only be avoided if the distribution uniquely specifies the best arm. With this constraint on the environment, our approach would already be competitive with the stronger baseline, because in this case the best arm is also identified by the best mean.

On the Movielens dataset, the unexpected ordering of the expected and sampled version is explained by the variance in the results (also note the errorbars bars). We re-ran both policies with 250 repetitions and obtained almost equal performance (no code changes).

In the crop experiment all approaches performed very similar due to an unfortunate choice in our setup. Namely, as context distribution we had chosen a Gaussian perturbation of a randomly chosen feature vector from the dataset (with variance informed from the dataset), and returned the chosen feature vector as context realization. With this setup, the context realization was always the mean of the observed distribution, which caused the different approaches to have very similar performance. We now changed the setup and center the context distribution around a fixed perturbation of the original vector, which is also a more realistic scenario. With this change, we observe a clear separation of the different configurations as shown on the right. The plots will be updated in the revised version of our paper, and we will add more discussion.

Top: Movielens with 250 repetitions. Bottom: Updated crop experiment.

**Reviewer #3**

We would like to emphasize that we identify a novel bandit setting that arguably covers many important applications. To someone familiar with the setting, our formulation leads to an analysis that might seem relatively straight forward, at least for the exact version of our algorithm. The sample-based version of our algorithm, however, poses additional technical challenges, as the sequential arm selection strategy can introduce bias into to estimated feature vectors. The bias carries on to the least squares regression and the regret, and needs to be controlled (Appendix A1 and A2, in particular eq (7), (11)-(12)). The improvement for the setting where the context realization is observed (Theorem 2) also requires a modification of the standard analysis (eq (17) and below). Finally, we provide a clean formulation of the kernelized setting, which we expect to be of interest for the Bayesian optimization community.

Lamprier, S., Gisselbrecht, T., and Gallinari, P. (2018). Profile-based bandit with unknown profiles. *The Journal of Machine Learning Research*, 19(1):2060–2099.


[Meta-Review · NeurIPS 2019]

This paper studies a linear bandit problem where the feature vectors of arms are unknown and drawn from known distributions. The change to the learning algorithm is relatively straightforward, LinUCB where the feature vector is the average feature vector. Then, if the optimal arm is the best arm on average with respect to the feature distribution, all algebra from linear bandits is expected to generalize and the authors indeed get similar results. This paper was discussed and all reviewers agree that it is a borderline. A closely related setting was studied in prior work. The algorithm and its analysis are not surprising and rather standard.